# Improved Mechanical and Corrosion Properties of Powder Metallurgy Austenitic, Ferritic, and Martensitic Stainless Steels by Liquid Phase Sintering

**DOI:** 10.3390/ma15165483

**Published:** 2022-08-09

**Authors:** Ming-Hsiang Ku, Lung-Chuan Tsao, Yu-Jin Tsai, Zih-Jie Lin, Ming-Wei Wu

**Affiliations:** 1Department of Materials and Mineral Resources Engineering, National Taipei University of Technology, No. 1, Sec. 3, Zhong-Xiao E. Rd., Taipei 10608, Taiwan; 2Institute of Materials Science and Engineering, National Taipei University of Technology, No. 1, Sec. 3, Zhong-Xiao E. Rd., Taipei 10608, Taiwan; 3Department of Materials Engineering, National Pingtung University of Science and Technology, No. 1, Shuefu Rd., Neipu, Pingtung 91201, Taiwan; 4Fusheng Precision Co., Ltd., No. 9, Xingzhong St., Taoyuan 33068, Taiwan

**Keywords:** powder metallurgy, stainless steel, sintering, hardness, corrosion

## Abstract

Powder metallurgy (PM) has been widely used to produce various steels in industry, mainly due to its capabilities for manufacturing nearly net-shaped products and mass production. To improve the performances of PM stainless steels, the roles of 0.6 wt% B additive in the microstructures, mechanical properties, and corrosion resistances of PM 304L austenitic, 410L ferritic, and 410 martensitic stainless steels were investigated. The results showed that adding 0.6 wt% B significantly improved the sintered densities of the three kinds of stainless steels due to the liquid phase sintering (LPS) phenomenon. The borides in 304L + 0.6B, 410L + 0.6B, and 410 + 0.6B were rich in B and Cr atoms but deficient in Fe, Ni, or C atoms, as analyzed by electron probe micro-analysis. Furthermore, the B additive contributed to the improved apparent hardness and corrosion resistance of PM stainless steels. In the 410L stainless steel, the 0.6 wt% B addition increased the corrosion voltage from −0.43 V_SCE_ to −0.24 V_SCE_ and reduced the corrosion current density from 2.27 × 10^−6^ A/cm^2^ to 1.93 × 10^−7^ A/cm^2^. The effects of several factors, namely: porosity; the generation of boride; the matrix/boride interfacial areas; Cr depletion; and the microstructure on the corrosion performances are discussed. The findings clearly indicate that porosity plays a predominant role in the corrosion resistances of PM austenitic, ferritic, and martensitic stainless steels.

## 1. Introduction

Powder metallurgy (PM) is a versatile means of producing nearly net-shaped metallic materials in industry. The biggest challenge of PM steel is the 5–15 vol% porosity, which obviously deteriorates the mechanical and corrosion performances [1,2,3]. Several techniques, such as warm compaction, double-press double-sintering, powder forging, and spark plasma sintering can decrease the porosity, though these techniques greatly raise the production cost [4,5,6,7,8]. Liquid phase sintering (LPS) is an economical technique to facilitate the sintering densification of PM materials by adding specific alloying elements for liquid generation without obviously increasing the cost [9,10,11,12,13,14,15,16]. For PM steel, boron (B) is an effective alloying element for LPS due to an Fe–B eutectic reaction [11,12,13,14,15,16].

The alloying elements of B-alloyed PM steel can modify the liquid generation temperature. In PM alloy steels, Cr and Mo raise the liquid formation temperature, but Ni decreases it [13,16,17,18]. However, 0.5 wt% C additive in B-alloyed PM steel not only induces the generation of a secondary liquid but also increases the equilibrium liquid volume at 1250 °C by ~5 vol% [17]. Moreover, the crystal structures of borides are also modified by the alloying elements. Several types of boride structures, such as M_2_B, M_3_(B,C), and M_23_(B,C)_6_, where M represents the metallic elements, have been experimentally identified in the literature [3,12,13,14,17,18]. Recently, You et al. [14] determined that MoFe(C,B) boride (WCoB-type) exists in the Fe-5Mo-0.8C-0.4B PM steel by using an atomic resolution high angle annular dark field (HAADF) scanning transmission electron microscope (STEM). In the Fe-0.5Mo-0.5C-0.4B PM alloy steel, increasing the sintering temperature from 1200 °C to 1250 °C changes the boride structure from M_2_B to M_3_(B,C) [17]. Thus, the role of alloying elements in the LPS and microstructure is very complicated. 

The mechanical properties of B-alloyed PM stainless steels have been investigated in various studies [2,3,12,15]. Wu et al. [3] studied the effects of B on the mechanical properties and fracture behaviors of PM 304L and found that 0.6 wt% B additive significantly increases the ultimate tensile strength by 63% without obvious sacrifices to tensile elongation. Pandya et al. [1] found that increasing the sintering temperature of PM 316L from 1200 °C to 1400 °C decreases the porosity and improves the corrosion resistance. However, the effects of B on the microstructure, particularly the boride, and the mechanical and corrosion properties of B-alloyed PM stainless steel with various matrices have not been clearly understood and compared to date. 

The 304L austenitic and 410L ferritic stainless steels are two major stainless steels which are extensively used in the PM industry. To understand the role of the matrix of B-alloyed PM stainless steel in the LPS and material performances, 410 martensitic stainless steel was also studied in this work. The purpose of this study was to investigate the influences of B on the microstructure, mechanical and corrosion performances of PM 304L austenitic, 410L ferritic, and 410 martensitic stainless steels. The microstructures were examined using an optical microscope (OM), a scanning electron microscope (SEM), and an electron probe micro-analyzer (EPMA). The apparent hardness of the material and boride microhardness were measured. The corrosion resistances were also analyzed using potentiodynamic polarization measurement. The results in this study show that the B additive was beneficial to the sintered density, apparent hardness, and corrosion resistance of the three PM stainless steels. 

## 2. Experimental Procedure

The sizes of the 304L, 410L, B, and graphite powders used in this study are shown in Table 1, as measured by laser light scattering particle analyzer (Mastersizer 3000E, Malvern Instruments LTD., Worcestershire, UK). Figure 1 shows the morphologies of the 304L, 410L, B, and graphite powders using SEM. The 304L and 410L powders were irregular in appearance. The B and graphite powders had granular and flaked shapes, respectively. The 0.6 wt% B powder was added to the 304L and 410L powders to produce 304L + 0.6B and 410L + 0.6B powder mixtures, respectively. To prepare the B-alloyed 410 martensitic stainless steel, 0.13 wt% graphite and 0.6 wt% B powders were added to the 410L powder to formulate the 410 + 0.6B powder mixture. Martensitic stainless steel powder is hard, so it has low compressibility and green density. Thus, the 410 PM martensitic stainless steel was produced by mixing 410L ferritic stainless steel powder with an adequate graphite powder. A lubricant of 0.75 wt% Acrawax was also mixed with the 304L, 410L, 304L + 0.6B, 410L + 0.6B, and 410 + 0.6B powder mixtures. 

These previous powder mixtures were uniaxially compressed at 600 MPa to prepare green disks with diameters of 13 mm and thicknesses of 5 mm. For debinding, the green specimens were heated at 5 °C/min to 550 °C in an Inconel tube furnace, held at that temperature for 15 min, and then furnace cooled to room temperature. The 304L, 304L + 0.6B, and 410 + 0.6B debound specimens were heated at 10 °C/min from room temperature to 1250 °C in a graphite vacuum furnace, held for 1 h, and then furnace cooled, as illustrated in Figure 2. To prevent the vaporization of Cr atoms, the vacuum pressure during sintering was maintained at 1–10 torr by argon refilling. To produce ferritic stainless steel, the 410L and 410L + 0.6B debound samples were heated at 10 °C/min from room temperature to 1250 °C in an Inconel tube furnace in hydrogen because C atoms in the residual C-containing gases of the graphite vacuum furnace could diffuse into 410L and transform the microstructure from ferrite to martensite.

The sintered densities of the sintered parts were measured using the Archimedes method. The sintered samples were cut with a grinding wheel cutting machine and then placed in a hot mounting apparatus. The metallographic samples were subsequently ground and polished with 1 μm and 0.3 μm aluminum oxide powder polishing solutions. After polishing, samples were placed in alcohol under an ultrasonic vibrator to remove the polishing fluid for 10 min before finally being etched with aqua regia. The microstructure was observed using an OM and an SEM (JSM-6510LV, JEOL, Tokyo, Japan). To calculate the porosity, the OM images were processed in image analysis software (Image-Pro Plus 7.0.1, Media Cybernetics Inc., Rockville, MD, USA). To clarify the elemental distributions in the various specimens after sintering, EPMA (JXA-8530F PLUS, JEOL, Japan) was used to investigate the elemental maps of Fe, Cr, Ni, B, and C atoms. 

The apparent hardnesses of the stainless steel samples were tested with a Rockwell hardness tester (8150SK, Indentec Hardness Testing Machines Ltd., West Midlands, UK) using the Rockwell B scale or Rockwell C scale. A Vickers microhardness testing machine (MMX-T, MATSUZAWA, Tokyo, Japan) with a loading of 5 gf was also utilized to measure the microhardnesses of the matrix and boride. The reported hardness values are the averages of five measurement points. To examine the corrosion resistances of various samples, a potentiostat (ECW-5000, JIEHAN Technology Co., Taichung, Taiwan) was used to measure the polarization curve by potentiodynamic polarization test at room temperature. The working, reference, and auxiliary electrodes were the sintered specimens, saturated calomel electrode (SCE), and platinum, respectively. The corrosion potential (E_corr_) and corrosion current density (I_corr_) were determined by Tafel plot. The test environment was a 3.5 wt% NaCl aqueous solution, and the scanning speed was 0.001 V/s. 

## 3. Results and Discussion

### 3.1. Microstructure

Figure 3 shows the microstructures of 304L and 304L+ 0.6B steels sintered at 1250 °C for 1 h in a vacuum. The matrix, pore, and eutectic area are indicated by arrows in Figure 3, Figure 4 and Figure 5. The results indicated that numerous pores remained in 304L after sintering, as shown in Figure 3a. Adding 0.6 wt% B to 304L obviously decreased the porosity, and the formation of eutectic areas was clearly observed, as shown in Figure 3b,c. The matrices of the 304L and 304L + 0.6B steels were austenite [3]. Figure 4 shows the microstructures of the 410L and 410L + 0.6B steels sintered at 1250 °C for 1 h in hydrogen. The matrices of the 410L and 410L + 0.6B were both ferrite. Many pores were observed in 410L; however, the porosity of 410L + 0.6B was obviously decreased, as can be seen from comparing Figure 4a,b. Eutectic areas were also found in Figure 4b,c. The microstructure of the 410 + 0.6B steel sintered at 1250 °C for 1 h in vacuum is shown in Figure 5. The results indicated that the matrix of 410 + 0.6B was martensite, and eutectic areas were also observed in Figure 5. 

According to Figure 3a and Figure 4a, the microstructures of 304L and 410L steels were typical of solid-state sintering, and thus numerous irregular pores were present in the two steels. However, the 0.6 wt% B addition in the 304L, 410L, and 410 steels facilitated the LPS by eutectic reaction and sufficiently reduced the porosity. The eutectic areas along the grain boundaries were clearly visible. LPS can be divided into three stages: particle rearrangement, solution–reprecipitation, and finally solid-state sintering [9]. The stage of particle rearrangement contributed to the rapid sintering densification and elimination of pores because the stacking status of the loosely-packed powder in the green part was greatly improved by the generation of liquid phase [9]. In the stage of solution–reprecipitation, the grain and pore sizes were obviously enlarged, and the shapes of the grains and pores were spheroidized [9]. As shown in Figure 3, Figure 4 and Figure 5, the porosities of the stainless steels were apparently reduced by adding 0.6 wt% B. Moreover, the matrix grains and pores were round in shape, and grain growth was obvious. The microstructures in Figure 3b, Figure 4b and Figure 5a were typical of those after the second stage of LPS sintering (solution–reprecipitation).

### 3.2. Sintered Density and Porosity

The green densities, sintered densities, and porosities after sintering of the five compositions are listed in Table 2 [3,19]. The green densities of 304L, 304L + 0.6B, 410L, 410L + 0.6B, and 410 + 0.6B were 6.48 g/cm^3^, 6.47 g/cm^3^, 6.51 g/cm^3^, 6.46 g/cm^3^, and 6.45 g/cm^3^, respectively. The sintered densities of 304L and 410L were 6.86 g/cm^3^ and 7.16 g/cm^3^, respectively. With the addition of 0.6 wt% B, the sintered densities of 304L + 0.6B, 410L + 0.6B, and 410 + 0.6B were 7.47 g/cm^3^, 7.64 g/cm^3^, and 7.58 g/cm^3^, respectively. In addition, the porosities of 304L and 410L were as high as 16.6 vol% and 8.9 vol%, respectively. However, the 0.6 wt% B additive significantly decreased the porosities of 304L + 0.6B, 410L + 0.6B, and 410 + 0.6B to no higher than 4.0 vol%. The results in Table 2 well corresponded to those in Figure 3, Figure 4 and Figure 5. 

### 3.3. Elemental Distribution

To clarify the elemental distributions of the matrix and boride, the 304L + 0.6B, 410L + 0.6B, and 410 + 0.6B were examined using EPMA and are shown in Figure 6, Figure 7 and Figure 8, which present secondary electron images (SEI) or backscattered electron images (BEI). Figure 6 shows that in 304L + 0.6B, Fe and Ni atoms were uniformly and richly distributed in the austenitic matrix but deficient in the eutectic areas. B and Cr atoms were mainly distributed in the boride of the eutectic area, with relatively few in the austenitic matrix. Figure 7 indicates that the ferritic matrix was rich in Fe and deficient in B and Cr. The boride in the eutectic area was rich in B and Cr. In 410 + 0.6B, the B and Cr atoms were also predominantly concentrated in the boride of the eutectic area, as shown in Figure 8. In contrast, Fe and C atoms were mainly concentrated in the matrix. The segregation of C in the matrix improved the hardenability, and thus the hardenability of the matrix was sufficiently high for transformation into martensite after sintering. Based on the above results, it is clear that B and Cr elements tended to jointly concentrate at the boride in the eutectic area, and other elements, including Fe, Ni, or C, were mainly distributed in the matrix. Such Cr segregation in the boride has been found in several B-alloyed PM stainless steels and alloy steels [3,12,15,18]. However, C atoms could be rich or deficient in the boride according to the crystal structures of boride [17].

Furthermore, Figure 6, Figure 7 and Figure 8 show that the morphologies of the boride and eutectic area were obviously divergent in the various stainless steels with the 0.6 wt% B addition. In 304L + 0.6B and 410L + 0.6B, the microstructures of the eutectic area were networked, and the boride appeared as particles or strips. However, in 410 + 0.6B, the eutectic area was not networked, and the boride was blocky. The size of the boride in 410 + 0.6B was significantly larger than those in 304L + 0.6B and 410L + 0.6B. The boride in 304L + 0.6B was smaller than 1 μm, as shown in Figure 3c, and thus the smallest among the three stainless steels with 0.6B. These previous findings clearly demonstrate that the morphology and boride size were obviously changed by the composition of the stainless steel and must be clearly identified in the future. 

### 3.4. Hardness 

The microhardnesses of the boride and matrix and the apparent hardnesses of the five stainless steels are listed in Table 3. The results showed that the 0.6 wt% B addition increased the microhardnesses of the matrices in 304L and 410L from 146 HV to 209 HV and from 107 HV to 132 HV, respectively. Moreover, the microhardness of the matrix in 410 + 0.6B was as high as 536 HV due to the martensitic matrix. The results in Table 3 and Figure 4 and Figure 5 clearly indicate that the addition of 0.13 wt% graphite powder fully transformed the matrix structure of 410L + 0.6B from ferrite to martensite. On the other hand, the microhardnesses of the borides in 410L + 0.6B and 410 + 0.6B were about 1620 HV. However, because the average boride in 304L + 0.6B was smaller than 1 μm in size, as shown in Figure 6, the microhardness could not be accurately measured by Vickers microhardness tester, even though the applied loading was merely 5 gf. The microhardness of the eutectic area in 304L + 0.6B was 450 HV. Furthermore, the apparent hardnesses of 304L and 410L were increased by 36 HRB and 30 HRB, respectively, by the addition of 0.6 wt% B due to the large decrease in porosity after LPS densification. The addition of 0.13 wt% graphite powder significantly increased the apparent hardness of 410L + 0.6B from 83 HRB to 48 HRC because the matrix changed from ferrite to martensite. 

The microhardnesses of several kinds of borides in B-alloyed PM and wrought steels have been investigated [17,20,21,22]. The microhardnesses of the borides in the Fe-0.5Mo-0.5C-0.4B, Fe-1.8Ni-0.5Mo-0.5C-0.4B, and Fe-4Ni-0.5Mo-0.5C-0.4B PM alloy steels were analyzed by Wu et al. [17]. They found that the microhardnesses of M_2_B boride and M_3_(B,C) borocarbide are 1562–1619 HV and 1023–1103 HV, respectively. In a previous study, the boride in 304L + 0.6B was identified as M_2_B boride by electron backscatter diffraction [3]. The elemental distributions of B and Cr atoms in 304L + 0.6B completely matched those in 410L + 0.6B and 410 + 0.6B. Carbon atoms were not concentrated in the boride of 410 + 0.6B. Moreover, the microhardnesses of the borides in 410L + 0.6B and 410 + 0.6B were about 1620 HV. Therefore, the borides of 410L + 0.6B and 410 + 0.6B should be M_2_B borides.

### 3.5. Corrosion Performance

Figure 9 and Table 4 show the potentiodynamic polarization curves and corrosion properties of the five specimens in 3.5 wt% NaCl aqueous solution. The polarization curves of the five types of steel in Figure 9 exhibited similar shape trends, and active regions were observed [23,24,25]. Table 4 shows that the corrosion potentials of 304L and 304L + 0.6B were −0.37 V_SCE_ and −0.28 V_SCE_, respectively. Moreover, the 0.6 wt% B additive decreased the corrosion current density of 304L from 4.07 × 10^−6^ A/cm^2^ to 2.13 × 10^−7^ A/cm^2^. These previous findings indicated that the corrosion resistance of 304L + 0.6B was better than that of 304L. Moreover, the B addition in 410L also significantly increased the corrosion potential from −0.43 V_SCE_ to −0.24 V_SCE_ and decreased the corrosion current density from 2.27 × 10^−6^ A/cm^2^ to 1.93 × 10^−7^ A/cm^2^, indicating the positive effect of 0.6 wt% B addition on the corrosion resistance of 410L. Figure 10 shows the as-polished surfaces of 304L and 304L + 0.6B after the corrosion tests. The results indicated that in 304L, large and irregular pores with lengths of about 100 μm formed during the corrosion test, and the surfaces were seriously corroded. In contrast, no large pores were observed in 304L + 0.6B after the corrosion test.

The preferential sites for corrosion were pores, grain boundaries, phase boundaries, and second phases, as reported in previous studies [25,26,27]. The B additive could possibly deteriorate the corrosion resistance due to the generation of boride and additional matrix/boride interfacial areas. Moreover, boride is rich in Cr, leading to Cr depletion in the matrices of PM stainless steels, as shown in Figure 6, Figure 7 and Figure 8, and impaired corrosion resistance [28].

According to the findings in Figure 9 and Figure 10 and Table 4, it is clear that the high porosity was the predominant factor in the corrosion resistances of PM 304L and 410L. High porosity obviously deteriorates the corrosion resistance because the presence of pores increases the effective surface area and inhibits passivation [26]. Decreasing the porosity of the PM stainless steels obviously reduced the effective areas exposed to the corrosion attack and improved the corrosion resistance.

In general, the corrosion resistance of austenitic stainless steels is higher than those of ferritic stainless steels, and that of martensitic stainless steel is the lowest. As shown in Table 4, the 410L + 0.6B sample exhibited the highest corrosion resistance, followed by the 304L + 0.6B and then 410 + 0.6B samples. Surprisingly, the corrosion resistance of 410L + 0.6B was higher than that of 304L + 0.6B. This phenomenon can be attributed to the lower porosity in the 410L + 0.6B, as shown in Table 2. The corrosion resistance of 410 + 0.6B was the lowest among the three stainless steels containing B because the matrix of 410 + 0.6B was martensite with a high C content. However, the corrosion resistance of 410 + 0.6B was even better than those of 304L and 410L. These findings clearly show that porosity plays the most important role in the corrosion resistance of PM stainless steels. The generation of boride, the matrix/boride phase boundaries, the Cr depletion, and the matrix microstructure had minor effects on the corrosion resistance. Consequently, adding 0.6 wt% B to PM austenitic, ferritic, and martensitic stainless steels could significantly improve their apparent hardnesses and corrosion resistances, mainly due to the decrease in porosity resulting from the LPS.

## 4. Conclusions

The influences of 0.6 wt% B additive on the microstructure, mechanical properties, and corrosion resistances of PM stainless steels were investigated. The results indicated that the 0.6 wt% B addition significantly facilitated LPS and reduced the porosity. The large decrease in porosity due to the addition of B contributed to the improved apparent hardnesses of the PM stainless steels.Eutectic areas and boride formed with the B addition, and EPMA elemental maps indicated that the boride in the 304L + 0.6B, 410L + 0.6B, and 410 + 0.6B PM stainless steels was rich in B and Cr.The results of potentiodynamic polarization tests clearly showed that the B addition also obviously improved the corrosion resistances of the 304L, 410L, and 410 PM stainless steels. Porosity plays a decisive role in determining the corrosion performances of PM stainless steels. The formation of boride, the matrix/boride phase boundaries, the Cr depletion, and the matrix microstructure did not dominate the corrosion resistance of the PM stainless steels investigated in this study.

## Figures and Tables

**Figure 1 materials-15-05483-f001:**
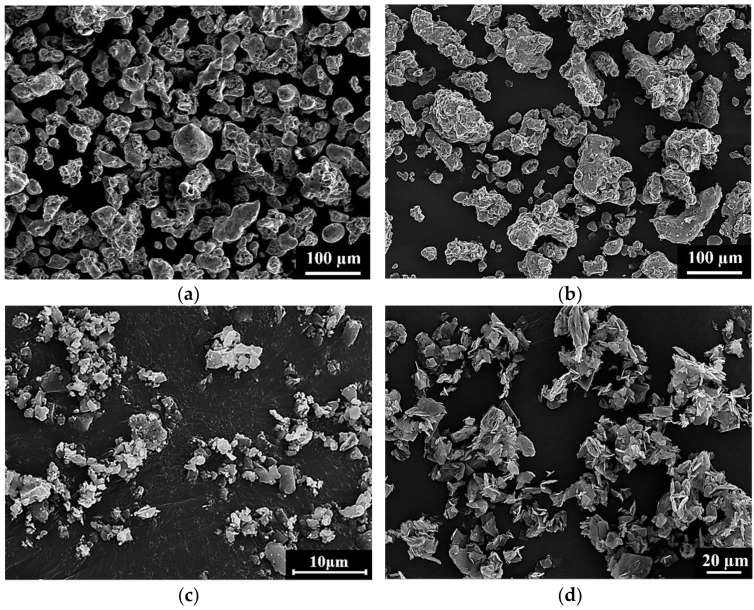
The morphologies of the raw powders used in this study. (**a**) 304L (**b**) 410L (**c**) B (**d**) graphite.

**Figure 2 materials-15-05483-f002:**
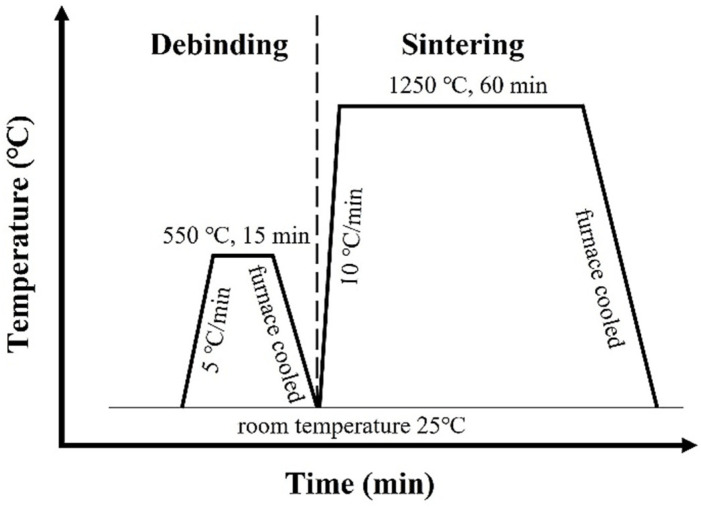
The schematic of the debinding and sintering schedule.

**Figure 3 materials-15-05483-f003:**
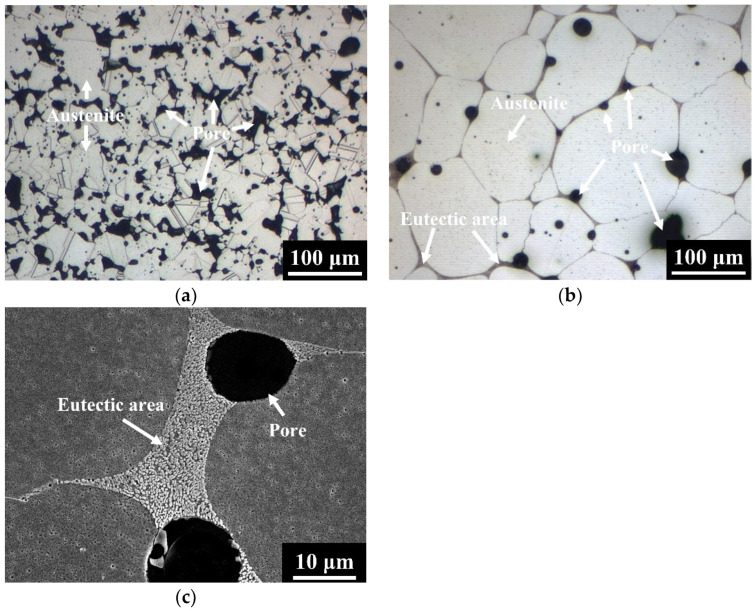
Microstructures of the (**a**) 304L and (**b**) 304L + 0.6B PM stainless steels. (**c**) High magnification of a eutectic area in 304L + 0.6B.

**Figure 4 materials-15-05483-f004:**
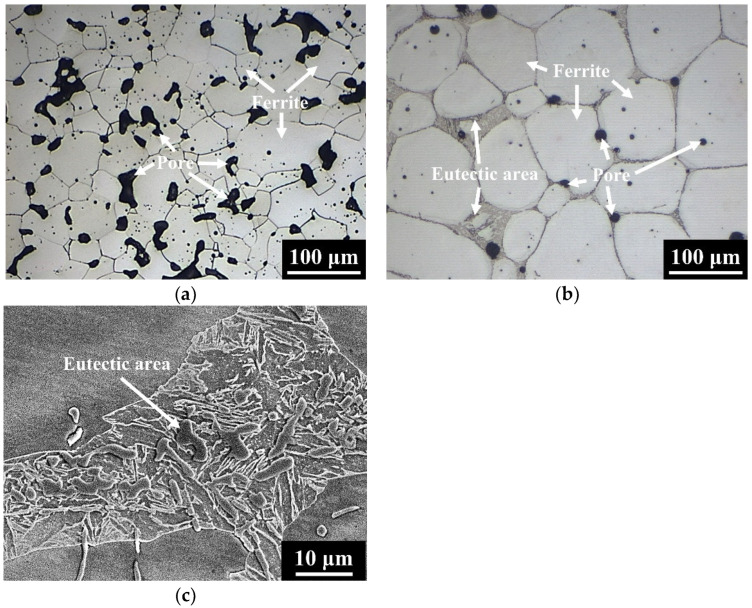
Microstructures of the (**a**) 410L and (**b**) 410L + 0.6B PM stainless steels. (**c**) High magnification of a eutectic area in 410L + 0.6B.

**Figure 5 materials-15-05483-f005:**
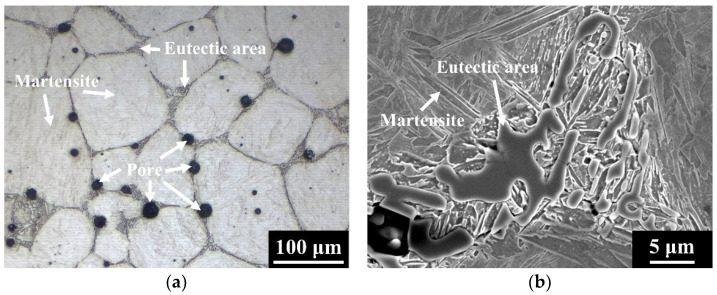
(**a**) Microstructure of the 410 + 0.6B PM stainless steel. (**b**) High magnification of a eutectic area in 410 + 0.6B.

**Figure 6 materials-15-05483-f006:**
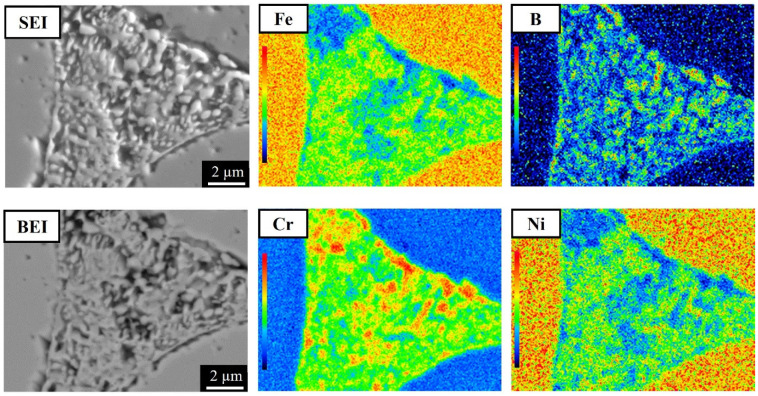
EPMA elemental distributions of 304L + 0.6B steel sintered at 1250 °C for 1 h in vacuum.

**Figure 7 materials-15-05483-f007:**
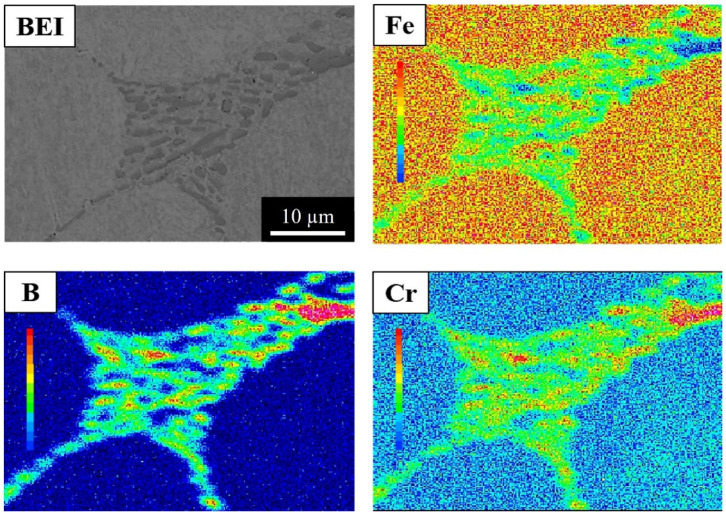
EPMA elemental distributions of 410L + 0.6B steel sintered at 1250 °C for 1 h in hydrogen.

**Figure 8 materials-15-05483-f008:**
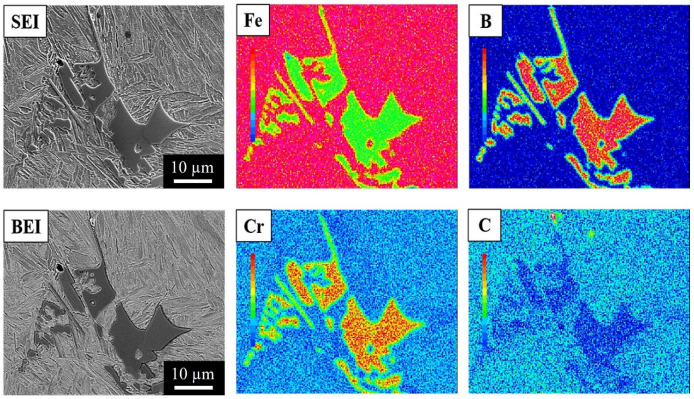
EPMA elemental distributions of 410 + 0.6B steel sintered at 1250 °C for 1 h in vacuum.

**Figure 9 materials-15-05483-f009:**
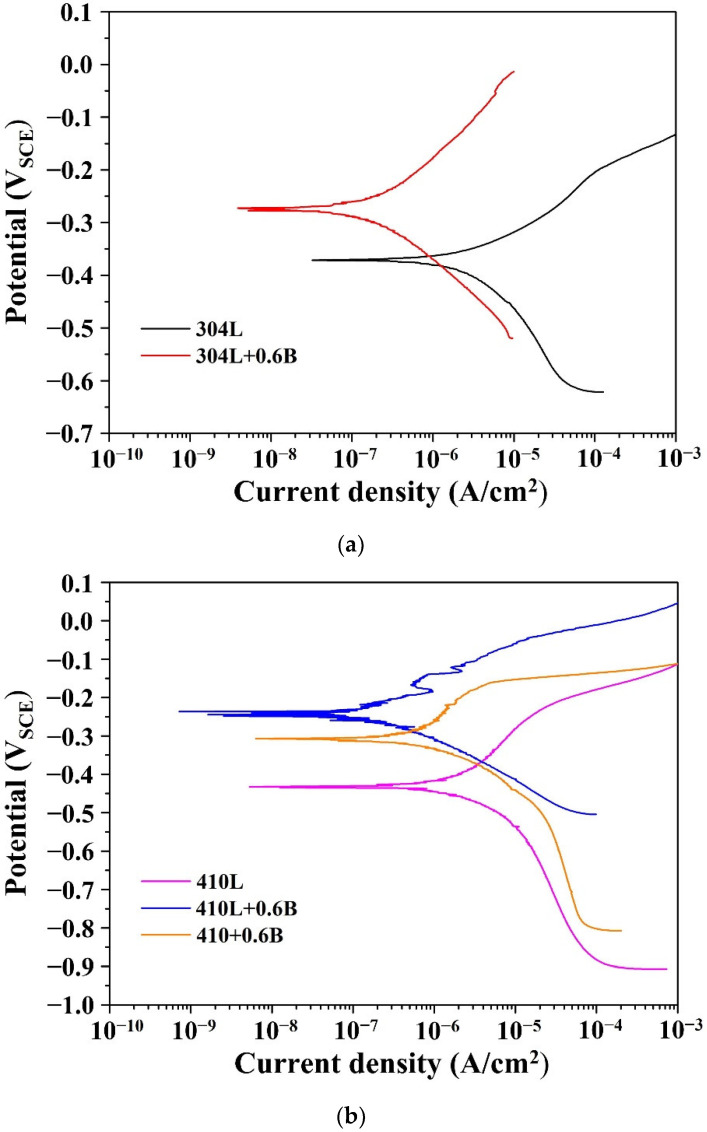
The polarization curves of PM stainless steels with and without B addition. (**a**) 304L and 304L + 0.6B (**b**) 410L, 410L + 0.6B, and 410 + 0.6B.

**Figure 10 materials-15-05483-f010:**
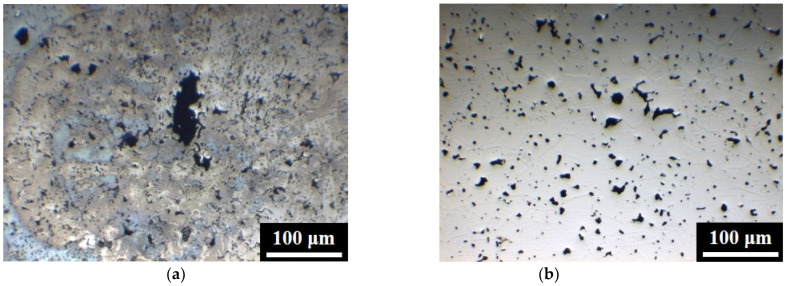
The surfaces of (**a**) 304L and (**b**) 304L + 0.6B after corrosion tests.

**Table 1 materials-15-05483-t001:** The sizes of 304L, 410L, B, and graphite powders used in this study.

Powder Type	D_10_ (µm)	D_50_ (µm)	D_90_ (µm)
304L	7.3	40.2	111
410L	4.9	39.8	108
B	0.8	2.1	5.6
Graphite	3.7	9.4	20.5

**Table 2 materials-15-05483-t002:** The green densities, sintered densities, and porosities after sintering of the five stainless steels investigated in this study.

Material	Green Density (g/cm^3^)	Sintered Density (g/cm^3^)	Porosity after Sintering (vol%)
304L [3]	6.48	6.86	16.6
304L + 0.6B [3]	6.47	7.47	4.0
410L [19]	6.51	7.16	8.9
410L + 0.6B [19]	6.46	7.64	1.4
410 + 0.6B [19]	6.45	7.58	3.4

**Table 3 materials-15-05483-t003:** The microhardnesses of boride and matrix and the apparent hardnesses of the five stainless steels.

Material	Microhardness of Matrix (HV)	Microhardness of Boride (HV)	Apparent Hardness (HRC or HRB)
304L [3]	146 ± 27	-	42 ± 1 HRB
304L + 0.6B [3]	209 ± 18	450 ± 33 (Eutectic area)	78 ± 1 HRB
410L	107 ± 3	-	53 ± 3 HRB
410L + 0.6B	132 ± 5	1621 ± 76	83 ± 2 HRB
410 + 0.6B	536 ± 6	1620 ± 62	48 ± 1 HRC

**Table 4 materials-15-05483-t004:** Corrosion properties of five PM stainless steels.

Material	Corrosion Current Density, I_corr_ (A/cm^2^)	Corrosion Potential, E_corr_ (V_SCE_)
304L	4.07 × 10^−6^	−0.37
304L + 0.6B	2.13 × 10^−7^	−0.28
410L	2.27 × 10^−6^	−0.43
410L + 0.6B	1.93 × 10^−7^	−0.24
410 + 0.6B	5.80 × 10^−7^	−0.31

## Data Availability

The raw data required to reproduce these results cannot be shared at this time as the data also forms part of an ongoing study.

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
