# Peer review of "Improved Mechanical and Corrosion Properties of Powder Metallurgy Austenitic, Ferritic, and Martensitic Stainless Steels by Liquid Phase Sintering"

_materials, 2022, doi:10.3390/ma15165483_

Round 1

Reviewer 1 Report

The current manuscript investigates the mechanical and corrosion properties of adding B to different PM stainless steel alloys. The presented results are interesting. However, some significant issues should be considered as follows:

 - The abstract should be reduced to only one paragraph, that displays the importance, problem statement, and main contribution of the current study.

- Power characterization should be presented in more detail, such as particle size distribution of the used material powders should be displayed.

- There is no data on the graphite powder characterization. 

- It is recommended to display the cycle chart of the green part heat treatment in a separate figure. 

- What are the repetitions of microhardness measurements for each sample?

- For figures 1, 2, and 3. all images should be described through the manuscript text. Also, there should indication of the pores and grains description in each figure image. 

- The elemental distribution in figures 3, 4, and 6 should be described in more detail through the manuscript text. 

- The standard deviation of the microhardness values should be added to Table 2. 

- The results and discussion section lacks more discussion and references to justify the presented results. 

- A bullet points style is highly recommended to arrange the conclusion section. The conclusion should focus on the main results, significant contribution, and novelty of the current work.   

Reviewer 2 Report

The present manuscript investigated the effect of boron addition on the sintering behavior leading to corrosion resistance. The manuscript is well-written and exciting. It can be accepted after minor revision.

1.       The abstract is too large and distracting. Please rewrite the abstract.

2.       Identify the phase constituents like porosity, eutectic phase, etc., by arrows.

3.       The given result in Table 1 is cited references means not from this work. So, please add the corresponding values for this study.

4.       The corrosion curves need to explain adequately. For example, why were there deviations (not smooth) in the B-added samples?

Reviewer 3 Report

The authors present a work entitled "Improved mechanical and corrosion properties of powder metallurgy austenitic, ferritic, and martensitic stainless steels by liquid phase sintering." The roles of 0.6 wt% B additive in the microstructures, mechanical properties, and corrosion resistance of PM 304L austenitic, 410L ferritic, and 410 martensitic stainless steels were investigated. This work is interesting, but some improvements should be made before it can be accepted for publication.

1. Try to shorten the Abstract section. Please present the core findings instead of stacking experimental results.

2. Introduction line 42. Spark plasma sintering is also a vital powder metallurgy process. Recent studies have found that using amorphous powder as a precursor can significantly promote densification. More importantly, it produces ultrafine-grained microstructures. Please add some discussions about the spark plasma sintering of amorphous powders. Here is some literature that can help with this process.

[1] Ding, Huaping, et al. "Enhancing strength-ductility synergy in an ex-situ Zr-based metallic glass composite via nanocrystal formation within high-entropy alloy particles." Materials & design 210 (2021): 110108.

3. Introduction section, line 67. Please add some introduction about why the authors chose the three steels. Furthermore, the final part of the introduction, no more than 100 words, should be added to briefly introduce what you did, how you did it, and what the core finding of this paper is. So, very much like a pared-down abstract, which eases readers into the results section that follows.

4. Figure 2, the scale bar is not displayed completely. Please check

5. Table 1, please use a three-line table. Table 2 and Table 3 do the same.

6. More discussions in depth are needed. For example, densification mechanisms during the sintering process, corrosion mechanisms, etc.

Round 2

Reviewer 1 Report

The review comments and recommendations are well addressed in the revised version. 

Reviewer 3 Report

The revision is OK. Please accept it.